The predatory bug Orius strigicollis shows a preference for egg-laying sites based on plant topography

Yu Chendi 1
Huang Jun junhuang1981@aliyun.com 1
Ren Xiaoyun 1
Fernández-Grandon G Mandela 2
Li Xiaowei 1
Hafeez Muhammad 1
Lu Yaobin luybcn@163.com 1
1 State Key Laboratory for Managing Biotic and Chemical Threats to the Quality and Safety of Agro-Products, Institute of Plant Protection and Microbiology, Zhejiang Academy of Agricultural Sciences , Hangzhou , Zhejiang , China
2 University of Greenwich, Natural Resources Institute , Chatham Maritime , Kent , UK
Gillespie Joseph
Electronic publication date: 2021 Jul 21
Publication date: 2021
Volume: 9
Electronic Location ID: e11818
Received 2021 Mar 16; Accepted 2021 Jun 29
Copyright: ©2021 Yu et al.
Copyright year: 2021
Copyright holder: Yu et al.
License: This is an open access article distributed under the terms of the Creative Commons Attribution License, which permits unrestricted use, distribution, reproduction and adaptation in any medium and for any purpose provided that it is properly attributed. For attribution, the original author(s), title, publication source (PeerJ) and either DOI or URL of the article must be cited.
License URL: https://creativecommons.org/licenses/by/4.0/

Keywords: Oviposition behavior, Site selection preference, Egg hatching, Plant topography, Orius strigicollis

Funding: Project National Key Research and Development 2017YFD0201000 National Natural Science Foundation of China 31772234 31801801 31672031 This study was supported by the Project National Key Research and Development (2017YFD0201000), and the National Natural Science Foundation of China (31772234, 31801801, 31672031). The funders had no role in study design, data collection and analysis, decision to publish, or preparation of the manuscript.

==============================
Background

Oviposition site selection is an important factor in determining the success of insect populations. Orius spp. are widely used in the biological control of a wide range of soft-bodied insect pests such as thrips, aphids, and mites. Orius strigicollis (Heteroptera: Anthocoridae) is the dominant Orius species in southern China; however, what factor drives its selection of an oviposition site after mating currently remains unknown.

Methods

Here, kidney bean pods (KBPs) were chosen as the oviposition substrate, and choice and nonchoice experiments were conducted to determine the preferences concerning oviposition sites on the KBPs of O. strigicollis. The mechanism of oviposition behavior was revealed through observation and measurement of oviposition action, the egg hatching rate, and the oviposition time.

Results

We found that O. strigicollis preferred the seams of the pods for oviposition, especially the seams at the tips of the KBPs. Choice and nonchoice experiments showed that females did not lay eggs when the KBP tail parts were unavailable. The rates of egg hatching on different KBP parts were not significantly different, but the time required for females to lay eggs on the tip seam was significantly lower. Decreased oviposition time is achieved on the tip seam because the insect can exploit support points found there and gain leverage for insertion of the ovipositor.

Discussion

The preferences for oviposition sites of O. strigicollis are significantly influenced by the topography of the KBP surface. Revealing such behavior and mechanisms will provide an important scientific basis for the mass rearing of predatory bugs.

Introduction

Insects tend to have the ability to select particular egg-laying sites to increase the survival rate of their offspring (Grostal & Dicke, 1999; Choh & Takabayashi, 2007; Barbosa-Andrade et al., 2019). Several factors can influence this behavior, for example, the existence of natural enemies or competitors (Rouault, Battisti & Roques, 2007; Choh, Sabelis & Janssen, 2015; Saitoh & Choh, 2018) and site properties such as food resource availability (Bond et al., 2005), illumination intensity (Yang, 2006) and temperature (Notter-Hausmann & Dorn, 2010). Apart from those common factors, some rare external physical factors such as the site size (Reich & Downes, 2003), shape, or color (Markheiser et al., 2017) can also play a role in the selection of oviposition sites.

The underlying cues mentioned above are complex and are less well understood than other aspects of insect behavior (Lundgen, Fergen & Riedell, 2008). However, due to the feeding habits of phytophagous insects, plants bring importance to the life histories and agricultural value of the predators that feed on these phytophagous insects (Lundgen, Fergen & Riedell, 2008; Puysseleyr & Hofte, 2011). Previous studies have shown that both plant species and variations in plant parts or tissues influence the oviposition behavior of predatory insects, (Isenhour & Yeargan, 1982; Coll, 1996; Lundgren & Fergen, 2006; Pascua et al., 2019). Of the many plant morphological features, the plant physical structure is one of the important factors that is known to significantly affect this reproductive behavior, either positively (Benedict, Leigh & Hyer, 1983; Griffen & Yeargan, 2002) or negatively (Simmons & Gurr, 2004). The mechanisms that drive female oviposition decisions have evolved such that female insects will choose sites with the optimal plant-based resources for the survival of their offspring (Malheiro et al., 2018; Mitchell et al., 2019). However, whether other factors influence the choice of oviposition sites by predatory insects remains to be explored.

Orius spp. are widely used in biological control methods to control many pests worldwide because they exhibit a higher search efficiency for their host than other species and are fast-moving and active (Minks, Harrewijg & Helle, 1989). For example, Orius strigicollis Poppius (Heteroptera: Anthocoridae) is an important native natural predator of a wide range of soft-bodied insect pests such as thrips, aphids, and mites in several agronomic systems (Cocuzza et al., 1997; Sengonca, Ahmadi & Blaeser, 2008; Zhang et al., 2012; Bonte & De Clercq, 2011) and feeds on lepidopteran pest eggs and young larvae (Bonte & De Clercq, 2011; Ali et al., 2020). There are several studies about O. strigicollis behavior that focus on its predatory advantages and its influence on agriculture (Zhou et al., 2006; Ali et al., 2020), but the mechanisms that influence the oviposition behavior of O. strigicollis based on plant characteristics remain poorly understood. However, studies on another zoophytophagous heteropteran, Orius insidiosus (Say), have reported that plant species, as well as the variations within each plant, significantly influence their oviposition behavior (Coll, 1996; Lundgren & Fergen, 2006; Pascua et al., 2019) and that they prefer to lay eggs on thinner epidermal plant surfaces, where the vesicular and cellular tissues are conducive to the survival and development of nymphs (Lundgen, Fergen & Riedell, 2008). As O. strigicollis is a natural enemy of plant pests, studying its oviposition site selection behavior will be very useful for the elaboration of mass rearing protocols.

Kidney bean pods (hereafter KBPs) are widely used in the indoor rearing of thrips and omnivorous bugs because of their freshness and convenience (Bonte & De Clercq, 2010; Li et al., 2018). We observed that O. strigicollis preferred KBPs for oviposition and seemed to have a preference for laying eggs on different parts of the KBPs. Therefore, we used KBPs as an oviposition substrate to study the mechanism of egg-laying selection preference in O. strigicollis. Thus, we hypothesized that there are differences in the number of eggs laid by O. strigicollis females in the different parts of the KBPs and that the most likely mechanism driving female oviposition decisions is the physical comfort of the laying position, which is directly related to the egg-laying efficiency. Here, we attempted to answer the following questions. (1) Do O. strigicollis females exhibit oviposition site selection behavior, and where do females choose to lay eggs? (2) Does the presence of the bean tail influence oviposition behaviors under choice and nonchoice conditions, or is the bean tip the best place for O. strigicollis females to lay eggs? (3) Why do O. strigicollis females select a specific location?

Materials & Methods

Insect rearing and experimental preparation

Orius strigicollis adults were collected from open areas and vegetable fields outside of Hangzhou (30.43898°N, 120.41134°E), Zhejiang Province, P.R. China, and maintained in a climate-controlled room. The rearing conditions were 26 ± 2 °C and 70 ± 10% RH, with a photophase of 14 h. All growth stages of O. strigicollis were reared in 4.3 L glass jars (see Supplemental Information 1 for more details) with a circular slant (i.e., the opening of the jar was on the side) capped with plastic screw-on lids. KBPs (length: 20.6 ± 5.1 cm) were used in the experiments as oviposition substrates for O. strigicollis. From the nymph to adult stages, the predatory bugs were fed western flower thrips, Frankliniella occidentalis (Pergande).

Oviposition site selection preferences

A pair of KBPs was laid flat on a filter paper inside the jars, and five mated O. strigicollis females were placed into each jar and allowed to oviposit for 48 h. Approximately 100 F. occidentalis nymphs were placed in each jar as food every day. The climatic conditions were the same as those described above. The KBPs were collected 48 h later for egg counting. The number of eggs per pod and the number of eggs in different positions on the pod (face or seam) were counted under a Nikon SMZ1500 zoom stereomicroscope (Nikon, Japan). The number of eggs was determined by counting the exposed opercula. The KBPs were divided into three parts for this count, i.e., the tail, middle, and head (Fig. 1A). A pair of KBPs was regarded as one separate biological replicate, with 20 replicates in total.

Figure 1 The preference of oviposition sites on kidney bean pods (KBPs) in Orius strigicollis.

(A) The percentage of eggs laid on the seam and face of the KBPs. Each bar represents the mean + SEM (N = 20). Asterisks (**) indicate a significant difference (P < 0.0001, the nonparametric Wilcoxon’s matched pairs test). (B) The percentage of eggs laid in the head, middle and tail parts from the seam of KBPs. Different letters indicate significant differences (P < 0.05, the nonparametric Friedman’s test). (C) Comparing of the mean number of eggs (+ SEM) laid on treatment (tail covered or restricting access) and the control (uncovered) by nonchoices assay (Ncontrol = 20, Ntreatment = 19). Asterisks (**) indicate significant differences (P < 0.0001, Student’s t- test). (D) Comparing of the mean number of eggs (+ SEM) laid on treatment (tail covered or restricting access) and the control (uncovered) by choices assay (Ncontrol = 20, Ntreatment = 14). Asterisks (**) indicate significant differences (P < 0.0001, Student’s t- test). (E) Comparison of the percentage of eggs laid on the right and left sides of KBP tail and middle sections. Each bar represents the mean + SEM (N = 20); different letters indicate significant differences (P < 0.05, the nonparametric Friedman’s test). (F) Comparison of the percentage of eggs laid on the tip and neck sections. Each bar represents the mean + SEM (N = 20); Asterisks (**) indicate significant differences (P < 0.0001, the nonparametric Wilcoxon’s matched pairs test).

Influence of restricting KBP access on the oviposition site selection in Orius strigicollis

The tail of the KBP was wrapped with parafilm to render the preferred oviposition site inaccessible, and then nonchoice and choice testing were conducted to determine the oviposition site selection of O. strigicollis. Nonchoice testing was conducted with the tail covered, and choice testing was also performed with the tail covered on one pod and that of another pod presented uncovered. Egg counts were conducted as described above. In each test, a pair of intact KBPs inside a jar was used as a control. The nonchoice testing was replicated 19 times, and the choice testing was replicated 14 times.

Differences in the egg number and egg hatching rate on the middle and tail of KBPs

To better identify the optimal oviposition site of O. strigicollis, we redefined the middle as the left middle (Middle-L) and right middle (Middle-R), and the tail as the left tail (Tail-L) and right tail (Tail-R). The side to which the kidney bean tip turns was defined as the left side, and the other side was defined as the right side (Fig. 1E). The tail of the KBP was then further categorized into four parts, i.e., the left neck (Neck-L), right neck (Neck-R), left tip (Tip-L), and right tip (Tip-R). The narrowest part of the tip and the position where the tip begins to widen (extending ca. 1.0 cm towards the head) were defined as the “tip”, while the rest was the “neck” (Fig. 1F). Each pair of KBPs in each jar was considered a group, and the experiments were performed again as described above. The egg numbers on each of the further-divided parts (left or right, neck or tip) were counted, and the number of eggs hatched after 5 days was also recorded. If the operculum was opened and no dead nymph was found around the operculum, egg hatching was considered successful. Each treatment was replicated 20 times.

Observation of egg-laying behavior and analysis of oviposition efficiency

During the control experiments (i.e., a pair of intact KBPs presented for the oviposition test), the egg-laying movements of 15 females on the tail and middle sections were observed, and the entire egg-laying process was recorded using a micro-video recording system (HDR-SR11E, Sony, Japan). When the start of the egg-laying movement was observed, an electronic timer (Deli, China) was used to determine how long females took to lay one egg on the tail or middle section.

Statistical analysis

Microsoft Excel (version 16.39) was used to record the data. The analysis was conducted using Prism 8 (version 8.4.0) and SPSS (version 26.0). The nonparametric Wilcoxon’s matched pairs test was used to compare the differences in egg laid (%) between different positions (i.e., face and seam) or different sites (i.e., tip and neck). The nonparametric Friedman’s test was used to analyze the differences in egg laid (%) between different parts (i.e., head, middle, and tail) or subsections (tail-R, tail-L, middle-R, and middle-L) of the KBPs. A t-test was used to compare the total number of eggs and the egg-laying efficiency between treatments.

Results

Oviposition site selection preferences

A total of 97.9% of the eggs were laid on the seam of the KBPs, and only 2.1% of the eggs were laid on the face (Fig. 1A; Z = 3.9, df = 19, P < 0.0001). Moreover, significant differences in egg numbers were observed between different KBP parts. More eggs were laid on the seam of the tail and middle than on the seam of the head, and the highest percentage of eggs was laid on the tail (Fig. 1B; χ2 = 32.7, n = 20, df = 2, P < 0.0001), i.e., more than half of the total eggs (50.9%). Overall O. strigicollis females laid more eggs on the seam of the KBPs, specifically on the tails.

Influence of restricting KBP access on the oviposition site selection in Orius strigicollis

A nonchoice experiment was conducted with tail coverage. In this treatment, the total number of eggs on each pod was 57.7% lower than that in the control (Fig. 1C; 55.1 ± 3.0 vs. 130.2 ± 7.1 individuals, t = 11.2, df = 18, P < 0.0001). A choice assay was also performed with the tail covered for one pod and another pod presented uncovered. In this case, the mean number of eggs per replicate was 87.6 ± 6.73 individuals, which was also 32.7% lower than that found in the control (Fig. 1D; t = 5.7, df = 13, P < 0.0001). The data indicated that O. strigicollis females did not lay more eggs on other parts of the KBP when the tail parts were unavailable.

Differences in the egg number and egg hatching rate at different oviposition sites except for the head of the KBPs

The right tail section (Tail-R) was the most preferred by O. strigicollis for oviposition, followed by the left middle section (Middle-L) (Fig. 1E, χ2 = 47.0, n = 20, df = 3, P < 0.0001). Furthermore, it was shown that the tip section was the most preferred by O. strigicollis for laying eggs (Fig. 1F; Z = 3.9, df = 19, P < 0.0001). The egg hatching rates on different sections were not significantly different, and all were higher than 80% (Fig. 2A, χ2 = 3.4, n = 20, df = 4, P = 0.489). Therefore, we indicate that factors such as the hatching or survival of eggs may not influence the selection of oviposition sites.

Figure 2 Behavioral mechanisms of oviposition site selection on kidney bean pods (KBPs) in Orius strigicollis.

(A) Percentage of eggs hatched of eggs laid on five parts, i. e., tip, neck, right middle, left middle, and left tail. Same letters indicate no significant differences (P > 0.05, the nonparametric Friedman’s test). Each bar represents the mean + SEM (N = 20). (B) The time spent (seconds) for females to lay one egg on the right tip and left middle sections. Each bar represents the mean + SEM (N = 15). Asterisks (**) indicate significant differences (P < 0.0001, Student’s t- test). (C/D) Photograph of female adult ready to lay eggs on the seam of the middle and tip sections.

Observation of egg-laying behavior and efficiency analysis

The average time that each female spent laying one egg on the tip was 28.7% shorter than the time spent to lay an egg on the middle section (Fig. 2B, t = 6.0, df = 14, P < 0.0001). We observed that laying eggs on the right tip seam was more efficient than laying eggs in another section (Figs. 2C/2D), which indicates that the more uneven the plant surface is, the more conducive it is to O. strigicollis oviposition.

Discussion

Postmating behavior such as oviposition site selection is observed in many insect species and is important for the reproduction of these species (Thompson, 1988). For example, Gryllus texensis Cade and Otte and some myrmecophilous butterfly species choose a suitable oviposition site for the survival of their offspring (Stahlschmidt & Adamo, 2013). In this study, we found that O. strigicollis females selected the seam of the KBPs rather than the face for egg laying. Such a preference difference for a different site on the same type of tissue or unit is common in oviposition site selection. For example, the lepidopteran multivoltine leafminers Phyllocnistis sp. prefer to lay eggs on only the lower-surface epidermal layer of the primary shoots, switching to lamma shoots when they appear later in the season (Ayabe, Minoura & Hijii, 2017). The longhorn beetle, Glenea cantor (F.), preferentially selects the upper section of kapok trees first for oviposition according to the bark moisture content from the top to the bottom of the trees (Lu et al., 2011). For insects with endophytic oviposition, the effects of characteristics of the plant tissues on their oviposition preference are more obvious (Lundgen, Fergen & Riedell, 2008). For example, Pascua et al. (2019) found that the strawberry calyx and flower petiole received more eggs than the other structures, and the thickness of the external tissues did not affect the oviposition of O. insidiosus. Likewise, Isenhour & Yeargan (1982) recorded a greater number of O. insidiosus eggs in the petiole of soybean flowers than in the other structures of the plant. Additionally, in our study, we found more than half of O. strigicollis eggs in the tip part, which was the primary oviposition site compared with other parts of the KBP. Further experiments were conducted to elucidate the hierarchy of preference of egg-laying females and identify the factors that influence it.

The results of the choice experiments suggest that first, when one of the preferred parts was unavailable, the total number of eggs laid decreased. Second, when none of the preferred parts were available the number of total eggs laid decreased rapidly, and although the left middle seam remained available for oviposition this site did not replace the preferred site. A previous study on mosquitoes suggested that the decreased oviposition rate observed on highly enriched leaves may be due to a pungent odor that is caused by the extreme anoxic environment and repels gravid female mosquitoes (Hoekman et al., 2007). Similar behavior was also observed in peach twig borers, Anarsia lineatella Zeller, of which female adults can determine whether peach fruits are fresh and viable for oviposition so that their larvae can have enough time to develop into adults before the peach fruits decompose (Sidney et al., 2008). In addition to odor and freshness, the sweetness and hardness of substrates also affected the egg-laying site selection of insects. (Wu et al., 2019) showed that activation of sweet neurons by sucrose can promote Drosophila females to become indifferent between two substrates of different hardness levels during egg laying. In this study, we assumed that the seam of KBP might be a good place since the eggs would remain more hidden, allowing it to provide a kind of refuge and that in the extreme (tip), the tissues may be softer than in the middle. This could also be related to oviposition efficiency.

However, regardless of the reason for the egg-laying site selection, insects choose to lay eggs on well-nourished hosts or tissue to ensure the healthy development and survival of their offspring (Jeong et al., 2016; Malheiro et al., 2018; Mitchell et al., 2019). Here, we found that O. strigicollis laid the most eggs at the tip of the KBPs, which indicated this location as their preferred oviposition site. The egg hatching rate is an important biological index used to measure host fitness or the suitability of oviposition substrates (Murai, Naraim & Sugiura, 2001; Bonte & De Clercq, 2011; Bonte & De Clercq, 2010; Krug & Sosa, 2019), and it is also the most intuitive criterion to judge (Castane & Zalom, 1994). Therefore, we further analyzed the hatching rates of eggs laid on different parts of the KBPs (tail vs. middle). The data showed that the hatching rates on these four sections (middle L and R, tail L and R) were not significantly different. We suggest that the factors that influence the selection of oviposition sites may not be those that restrict the hatching or survival of eggs. Additionally, we found that the eggs were embedded in the KBP tissue, and the lid of the egg was opened when it hatched. Embedding eggs may simply protect the eggs from predation or parasitism and from abiotic factors, in addition to stabilizing the eggs or keeping them in a moist environment (Shapiro & Ferkovich, 2006). According to the optimal oviposition theory, plant tissue acceptability for oviposition may also be affected by the subsequent development or survival of Orius nymphs (Jaenike, 1978; Lundgren & Fergen, 2006). It is known that in many Orius species newly emerged nymphs use plant tissues to gain energy and begin their dispersal. Perhaps in the tip part of the KBPs, the tissues are softer or have some characteristics that make it easier for the small nymphs to feed more easily, but this hypothesis is worth further systematic study.

Based on our observation of the entire egg-laying process of the females and our measurements of the time required for the females to lay eggs, we suggest that females select the tip of the KBPs as their first oviposition site to achieve higher egg-laying efficiency, and reducing their time spent ovipositing also reduces their risk of predation and allows more time for foraging and perching (Martens, 2001; Philippe, Besnard & Natalia, 2015). Furthermore, we suggest that the increased egg-laying efficiency is due to the ‘ergonomics’ of this egg-laying position. The females must use force to insert their eggs into the KBP. To achieve this, they need anchor points for both their propodeum and metapodium to push against to gain the required power. Comparing the seam at the tip and in the middle section of the KBPs, the females were able to clasp the tip of the KBPs using their propodeum. This allowed oviposition in KBPs with much greater ease (see Supplemental Information 2 for more details). In contrast, because the side of the KBP is nearly flat, the females are required to use more strength and expend more energy to insert their eggs there. There is a similar explanation for the low egg distribution on the seam on the other side—compared with the preferred seam, the other seam is relatively shallow, and more energy might be required for the females to lay their eggs inside it. Similar observations and speculations were also mentioned briefly by Shapiro & Ferkovich (2006), who speculated that female adults of O. insidiosus may need to take advantage of the internal angles or surface irregularities to gain leverage for the ovipositor.

Conclusions

In this study, according to different positions and parts, the KBPs (oviposition substrates) were divided into different sites, e.g., face, seam, tail (including the neck and tip), middle, and head. We found that the physical features of each site were ultimately reflected in the corresponding egg-laying efficiency. The results suggest that the preference for oviposition sites of O. strigicollis is significantly influenced by the topography of the KBP surface, and the more ‘comfortable’ the females are, the higher their egg-laying efficiency. The behavioral mechanism of the preference of O. strigicollis females for oviposition sites on the KBPs was found and identified, which is conducive to our later development of artificial media to attract O. strigicollis to lay eggs. The results also provide the necessary knowledge to advance the massive production of O. strigicollis for their release within the framework of an augmentative biological control strategy.

Supplemental Information

Supplemental Information 1 Photo of the feeding device used for all experiments

The diameter of the plastic screw-on lid is 13 cm, and the diameter of the central hole is nine cm. The central hole is covered with 100 mesh gauze. The filter paper is 20 cm length with 20 cm width. The total volume of mixed vermiculite (ca. 350 g) and sawdust (ca. 20 g) is 2 ± 0.5 L.

Click here for additional data file.

Supplemental Information 2 The oviposition behavior of Orius strigicollis

Comparing the seam at the tip and in the middle section of the KBPs, the females were able to clasp the tip of the KBPs using their propodeum. This allowed oviposition in KBPs with much greater ease. In contrast, because the side of the KBP is nearly flat, the females are required to use more strength and expend more energy to insert their eggs there.

Click here for additional data file.

Supplemental Information 3 Dataset for figures

Click here for additional data file.

Additional Information and Declarations

Competing Interests

Author Contributions

Data Availability

The authors declare there are no competing interests.

Chendi Yu performed the experiments, analyzed the data, prepared figures and/or tables, and approved the final draft.

Jun Huang conceived and designed the experiments, performed the experiments, analyzed the data, prepared figures and/or tables, and approved the final draft.

Xiaoyun Ren performed the experiments, authored or reviewed drafts of the paper, and approved the final draft.

G Mandela Fernández-Grandon and Xiaowei Li analyzed the data, prepared figures and/or tables, authored or reviewed drafts of the paper, and approved the final draft.

Muhammad Hafeez analyzed the data, authored or reviewed drafts of the paper, and approved the final draft.

Yaobin Lu conceived and designed the experiments, prepared figures and/or tables, authored or reviewed drafts of the paper, and approved the final draft.

The following information was supplied regarding data availability:

The raw measurements are available in the Supplemental File.

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
