# Peer review of "The predatory bug Orius strigicollis shows a preference for egg-laying sites based on plant topography"

_PeerJ, doi:10.7717/peerj.11818_

## Round 0.1 · original submission · Major Revisions

Dear Dr. Yu and colleagues:

Thanks for submitting your manuscript to PeerJ. I have now received three independent reviews of your work, and as you will see, the reviewers raised some concerns about the research. Despite this, these reviewers are optimistic about your work and the potential impact it will have on research studying the biology of egg-laying in anthocorids. Thus, I encourage you to revise your manuscript, accordingly, taking into account all of the concerns raised by all three reviewers.

While the concerns of the reviewers are relatively minor, this is a major revision to ensure that the original reviewers have a chance to evaluate your responses to their concerns. There are many suggestions, which I am sure will greatly improve your manuscript once addressed.

Please revise your experimental design for clarity. Your methods should be clearly outlined, and your experiments should be repeatable. There appear to be some important references missing from your citations.

Please also note that reviewer 1 has included a marked-up version of your manuscript.

Therefore, I am recommending that you revise your manuscript, accordingly, taking into account all of the issues raised by the reviewers.

I look forward to seeing your revision, and thanks again for submitting your work to PeerJ.

Good luck with your revision,

-joe

·

Basic reporting

The manuscript is, in general, well written, but I am not a native speaker. I think English needs to be checked before publishing as there are some phrases that are a bit confusing (at least for me).

The literature references are appropriate and sufficient, and the field background developed frames the proposed study.

The work is structured as a professional article, as well as the figures. In relation to these, there are some things to clarify / modify, which were indicated in the attached pdf version.
The raw data was shared as supplementary material.

In addition, there are some minor edit errors and related to the references that I detail in the pdf.

Experimental design

The study responds to the questions proposed, although the hypotheses are somewhat forced taking into account the theoretical framework described, so I suggest re-thinking the hypotheses or including some antecedent in the introduction (I detail this in the pdf version).

The methodology is described in sufficient detail, it may be convenient to add a figure to clarify the different parts of the bean that were taken into account in this study.

I have some dubts and comments about statistical analysis that I mention in the next item.

Validity of the findings

The raw data is provided, but I have some doubts about what data were used to carry out the analyzes, because the data provided and shown in the figures are in percentages, so the proposed statistic would not be the most appropriate, in that case perhaps it would be convenient to make a binomial logistic model.

On the other hand, I think the discussion could be improved. There are some redundant paragraphs that should be eliminated, and others in which some works are discussed that, in my opinion, are not very related to this work. I made more detailed observations in the pdf version.

The conclusions are connected to the original question investigated.

Additional comments

The work presented for publication is an interesting, well-structured work that explores the oviposition behavior of the hemipteran Orius strigicollis, and the results presented here will be very useful for the elaboration of mass rearing protocols.
I believe that these types of studies provide the necessary knowledge to be able to advance in the massive production of natural enemies for their release within the framework of the augmentative biological control strategy.

Reviewer 2 ·

Basic reporting

The manuscript analyzes how Predatory bugs show a preference for egg-laying sites based on plant topography. In my opinion the study is interesting, well written and presents interesting findings. However, I think that the introduction needs to be better contextualized. Authors needs to cite the results of other studies. It would strengthen the manuscpt. This is missing from the study.

Experimental design

The experimental design is ok.
Methods for collecting data are described with sufficient detail.

Validity of the findings

It is original primary research and results are interesting and should be published. Research questions are relevant and well defined but need little improvement.

Additional comments

Introduction needs improvement. Authors should cite the results of other studies related to this bug. And should highlight more clearly why this study is important and how it would help others for further research.

Reviewer 3 ·

Basic reporting

The manuscript by Yu et al. investigates the oviposition preference of the predatory bug Orius strigicollis on kidney bean pods. The authors conducted a series of simple, straightforward studies through choice and nonchoice experiments and video recordings. They found that O. strigicollis females prefer to lay eggs on seams of pods particularly the seams at the tips of the pods. They conclude that females prefer these areas because they achieve more leverage during ovipositor insertion.

Overall, the manuscript provides some interesting information on the oviposition preference of a predatory insect. However, I find that the information is very narrow in scope for a journal like PeerJ. I will leave it to the handing editor to decide if this manuscript fits with the scope of the journal. I would suggest the Journal of Insect Behavior instead.

Experimental design

I have several comments on the methods that should help improve the manuscript. Specifically, I find that the methods are not very clear and need some revisions as indicated in my comments to the authors. I also do not think the use of ANOVA is appropriate.

Validity of the findings

Although interesting, I do not think this manuscript is particularly novel. But, more critically, I do not think it is broad enough. The authors studied the oviposition preference of a single predatory bug. Thus, it has limited broad implications.

The hypothesis is based on the findings and not based on previous literature.

The statistical analysis needs to be revised since the data are not independent. Thus, I question whether the use of ANOVAs are appropriate.

Additional comments

The manuscript by Yu et al. investigates the oviposition preference of the predatory bug Orius strigicollis on kidney bean pods. The authors conducted a series of simple, straightforward studies through choice and nonchoice experiments and video recordings. They found that O. strigicollis females prefer to lay eggs on seams of pods particularly the seams at the tips of the pods. They conclude that females prefer these areas because they achieve more leverage during ovipositor insertion.

Overall, the manuscript provides some interesting information on the oviposition preference of a predatory insect. However, I find that the information is very narrow in scope for a journal like PeerJ. I will leave it to the handing editor to decide if this manuscript fits with the scope of the journal. I would suggest the Journal of Insect Behavior instead.

Besides, I have several comments that should help improve the manuscript. Specifically, I find that the methods are not very clear and need some revisions as suggested below. I also do not think the use of ANOVA is appropriate.

Specific comments:
The title is misleading since the authors only studies the oviposition preference on one predatory bug. I suggest “The predatory bug Orius strigicollis shows a preference for egg-laying sites based on plant topography”. Again, the authors cannot generalize to other predatory bug species and thus I feel this manuscript is narrow in scope for a journal like PeerJ.
Throughout the manuscript the authors use the words “behaviour” and “behavior”. Please be consistent based on journal guidelines.
Also throughout the manuscript, I prefer the use of “single-choice” and two-choice”. Instead of nonchoice and choice, respectively.
Line 75: It is unclear what the statement “These underlying cues” is referring to.
Line 89: delete “,” after “Anthocoridae),”
Lines 94-95: the sentence “the mechanism whereby the oviposition behavior of O. strigicollis are influenced by plant characteristics” is hard to read and should be revised.
Line 96: provide author name after scientific name
Line 100: change to “a natural enemy”
Line 107: the authors write that their hypothesis is that “females lay more eggs on the on the seams part of the KBPs, especially the bean tips”. It looks like this hypothesis was based on the results and not based on previous studies. It is unclear why they would expect this behavior. Are there specific plant characteristics and insect behaviors that would lead them to this hypothesis?
Line 122: provide author name after scientific name.
Line 125: how many thrips nymphs were provided?
Line 128: I suggest deleting “recoded. Finally, the egg numbers on different parts of the pods were”
Line 130: I am not clear what the authors call a “treatment”. This needs to be clearly defined.
Lines 131-136: This section needs to be revised. I am not clear what were the choices.
Line 134: what are the “treatments”?
Lines 134-135: So, a pair of pods wrapped in paraffin and a pair of pods without paraffin was provided inside a jar? What is the prediction?
Line 140: how did each site defined? Was each section marked?
Line 143: how did the authors determine egg hatching?
Line 146: which “control experiments”?
Line 152: I do not think the use of ANOVA is appropriate since the behaviors were not independent.
Lines 151-155: The authors had 2 pods of each “treatment” in the jars. These pods were not independent. Was the numbers polled or averaged?
Line 159: delete extra “(” after “face”
Line 165: “nonchoice”
Throughout the results, I would prefer to see percent increase numbers instead of words like “dramatically” or “significantly”
Line 166: change to “was xx% lower than”
Line 169: change to “was also xx% lower than”
Lines 175-176: this is a very confusing sentence. Maybe change to “The section of the right tip (Tip-R) was the most preferred by O. strigicollis for oviposition”
Line 181: change to “each female spent laying one”
Line 181: change to “was xx% shorter than”
Lines 183-184: change to “C/D), which indicates that”
Lines 184-185: change to “more conducive to O. strigicollis oviposition.”
Line 188: indicate author name after scientific name
Line 192: change to “For example, the lepidopteran”
Line 205: why was this “unexpectedly”
Line 211: the authors indicate “These examples” but it is unclear to me which examples the authors are referring to.
Line 244: The conclusions need to be a stand-alone section. However, the authors start the section stating: “The physical features of each site”. This is unclear. Which sites are they referring to?
Figure captions
I would delete the questions in the captions. They are confusing.
Figure 1. Delete “(How to choose “delivery room”?)” What is a delivery room? Not needed.
Figure 2. Delete “(Why do O. strigicollis choose this way?)” Again, not needed.
I do not think Figure 2A is needed since there was no difference among the sites. Just state in the text the average hatch rate +/- SE.
Figure 2B. Change the text in the y-axis to “Time spent for egg-laying”

---

## Round 0.2 · Minor Revisions

Dear Dr. Yu and colleagues:

Thanks for revising your manuscript. The reviewers are generally satisfied with your revision (as am I). Great! However, there are a few concerns to address. Please attend to these issues ASAP so we may move towards acceptance of your work.

Best,

-joe

·

Basic reporting

No comment

Experimental design

No comment

Validity of the findings

No comment

Additional comments

No comment

Reviewer 2 ·

Basic reporting

No comment

Experimental design

No comment

Validity of the findings

No comment

Additional comments

The revised version manuscript is better and can be accepted by the journal.

Reviewer 3 ·

Basic reporting

I had the opportunity to review an earlier version of this manuscript and I am satisfied with the way the authors have addressed my comments.

I only have a few additional minor edits:
Line 50: change to "for mass rearing of predatory bugs." I think it is worth being more specific.
Line 89: change to "and young larvae". See comment below.
Line 120: change to "on a filter"
Line 206: change to "of the KBP."
Line 221: I don't understand this sentence. Why the question? I think this is a mistake. Please correct.
Line 236: "newly emerged nymphs". To me, eggs hatch, nymphs emerge.
Lines 260-264. This is a long sentence. Please break into two sentences.

Experimental design

No comment

Validity of the findings

No comment

Additional comments

I had the opportunity to review an earlier version of this manuscript and I am satisfied with the way the authors have addressed my comments.

I only have a few additional minor edits:
Line 50: change to "for mass rearing of predatory bugs." I think it is worth being more specific.
Line 89: change to "and young larvae". See comment below.
Line 120: change to "on a filter"
Line 206: change to "of the KBP."
Line 221: I don't understand this sentence. Why the question? I think this is a mistake. Please correct.
Line 236: "newly emerged nymphs". To me, eggs hatch, nymphs emerge.
Lines 260-264. This is a long sentence. Please break into two sentences.

---

## Round 0.3 · accepted · Accept

Dear Dr. Yu and colleagues:

Thanks for revising your manuscript based on the concerns raised by the reviewers. I now believe that your manuscript is suitable for publication. Congratulations! I look forward to seeing this work in print, and I anticipate it being an important resource for groups studying the biology of egg-laying in anthocorids. Thanks again for choosing PeerJ to publish such important work.

Best,

-joe